# Sequential Flow Straightening for generative modeling

## Abstract

Even though the continuous-time generative models simulating ODEs and SDEs, such as diffusion models or flow-based models, have achieved great success in tasks such as large-scale image synthesis, generating high-quality samples from those models requires a large number of function evaluations (NFE) of neural networks. One key reason for the slow sampling speed of the ODE-based solvers that simulate these generative models is the high curvature of the ODE trajectory, which explodes the truncation error of the numerical solvers in the low-NFE regime. As straightening the probability flow is the key to fast sampling through the numerical solvers by increasing the tolerance of the solver, existing methods directly generate the joint distribution between the noise and data distribution and learn a linear path between those data pairs. However, this method also suffers from a high truncation error while generating the pair through the full simulation, thus worsening the sampling quality. To address this challenge, We propose a novel method called *sequential reflow*, a learning technique to straighten the flow that reduces the global truncation error and hence enabling acceleration and improving the synthesis quality. In both theoretical and empirical studies, we first observe the straightening property of our *sequential reflow*. Via sequential reflow, We achieved FID 3.97 with 8 function evaluations in CIFAR-10 dataset.

## 1 Introduction

In recent times, continuous-time generative models, exemplified by diffusion models (Song & Ermon, 2019; Song et al., 2021b; Ho et al., 2020) and flow-based models (Lipman et al., 2023; Liu et al., 2023a), have demonstrated significant improvement across diverse generative tasks, encompassing domains such as image generation (Dhariwal & Nichol, 2021), videos (Ho et al., 2022), and 3D scene representation (Luo & Hu, 2021), and molecular synthesis (Xu et al., 2022). Notably, these models have outperformed established counterparts like Generative Adversarial Networks (GANs) (Goodfellow et al., 2014) and Variational Autoencoders (VAEs) (Kingma & Welling, 2014). Operating within a continuous-time framework, these models acquire proficiency in discerning the time-reversal characteristics of stochastic processes extending from the data distribution to the Gaussian noise distribution in the

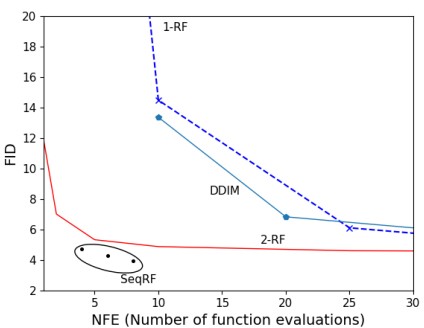

Figure 1: Generation Performance of SeqRF compared to existing results in CIFAR-10.

context of diffusion models. Alternatively, in the case of flow-based models, they directly learn the probability flow, effectively simulating the vector field.

Recently, Lipman et al. (2023) introduced a novel concept termed *flow matching* within continuous-time generative models. This approach focuses on learning the vector field connecting the Gaussian noise and the data distribution. The authors initially demonstrated that the marginal vector field, representing the relationship between these two distributions, is derived by marginalizing over the gradients of the conditional vector fields. Moreover, they found that the conditional vector field

yields optimal transport between the data and noise distributions, particularly when the noise distribution adheres to the standard Gaussian distribution. Learning the (marginal) vector field involves independent sampling from both the noise and data distributions, followed by the marginalization of the conditional vector field over the data distributions. Despite the advantageous property of identifying optimal transport paths with optimal coupling, this method faces challenges such as high learning variance and slow training speed attributable to the independent drawing of training data from data and noise distributions, which results in high gradient variance even at its convergence. This threatens the stability of the Ordinary Differential Equation (ODE) solver due to the accumulation of truncation errors. In response to this challenge, Liu et al. (2023a) proposed a method to straighten the trajectory by leveraging the joint distribution. This involves running the numerical ODE solver from the noise to approximate the data point. However, traversing a long path from noise to image space still leaves exploding global truncation errors unaddressed.

To address the challenge posed by truncation errors, we introduce a novel framework termed *sequential reflow*. This novel approach represents a straightforward and effective training technique for flow-based generative models, specifically designed to alleviate the truncation error issue. The key innovation lies in the segmentation of the ODE trajectory with respect to the time domain. In this strategy, we harness the joint distribution by partially traversing the solver, as opposed to acquiring the complete data with the entire trajectory. The overall concept of our method is briefly introduced in Figure 2.

Our contribution is summarized as follows.

- **Controlling the global truncation error**   We begin by highlighting the observation that the global truncation error of numerical ODE solvers experiences explosive growth, exhibiting superlinear escalation. This underscores the significance of generating the joint distribution from segmented time domains, thereby ensuring diminished global truncation errors through the strategic halting of the solver before the error reaches critical levels.

- **Main Proposal: Sequential Reflow for Flow Matching**   Our primary contribution is the introduction of *sequential reflow* as a method for flow matching in generative modeling. This innovative flow straightening technique aims to diminish the curvature of segmented time schedules. It achieves this by generating the joint distribution between data points at different time steps. Specifically, the source data point, originating from the training set enveloped in noise, undergoes the ODE solver constructed by pre-trained continuous-time generative models such as flow-based or diffusion models. This marks a departure from conventional reflow methods, which straighten the entire ODE trajectory.

- **Validation of Sequential Reflow**   We empirically validate that our sequential reflow method accelerates the sampling process, thereby enhancing the image synthesis quality; the sampling quality improves with the rectified flow, achieved by employing sequential reflow method subsequent to the initial flow matching procedure. Furthermore, We implement distillation on each time fragment, enabling the transversal of one time segment at a single function evaluation. This strategic approach results in superior performance, achieving a remarkable 3.97 Frechét Inception Distance (FID) with only 8 function evaluations on the CIFAR-10 dataset, as sketched in Figure 1.

The overall structure of our paper is as follows. In § 2, we lay the groundwork for our paper by introducing essential concepts such as rectified flow and elucidating on the truncation error associated with the ODE solver. In § 3, we present our primary contribution, the "sequential reflow" method. This approach effectively diminishes the curvature of the ODE trajectory, thereby enhancing the quality of the joint distribution for learning flow-based models. The section also encompasses theoretical and empirical validations, demonstrating the successful mitigation of the truncation error through our proposed method. § 4 delve into existing works that relate to our newly introduced framework. This contextualization provides a comprehensive understanding of the broader landscape within which our research unfolds. § 5 reports our experiments' findings, showcasing our proposed method's efficacy. We highlight improvements in image synthesis quality and accelerated sampling speed as key outcomes of our approach. Finally, in § 6, we summarize our contributions and suggest potential avenues for future research. This section serves as a reflection on the implications of our work and invites further exploration in related domains.

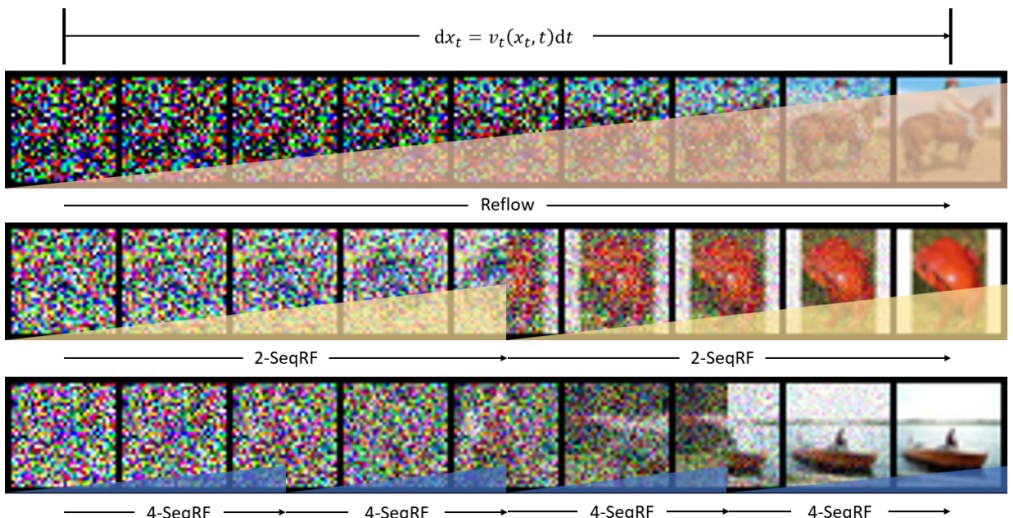

Figure 2: The concept figure of our method. The red, yellow and blue triangles represent the truncation error being accumulated in the corresponding time. Compared to the red reflow method, sequential reflow (SeqRF) mitigates marginal truncation error by running time-segmented ODE.

## 2  BACKGROUND: CONTINUOUS NORMALIZING FLOW AND FLOW MATCHING

Consider the problem of constructing the probability path between the two distributions $\pi_0$ and $\pi_1$ in the data space $\mathbb{R}^D$. Let the *time-dependent* vector field be $u : \mathbb{R}^D \times [0,1] \to \mathbb{R}^D$ with $t \in [0,1]$. Then the ODE generated by this vector field is

$$\mathrm{d}X_t = u(X_t, t)\mathrm{d}t. \tag{1}$$

Chen et al. (2018); Grathwohl et al. (2019) suggested the generative model called *Continuous Normalizing Flow (*CNF*)* that reshapes the simple, easy-to-sample density (i.e., Gaussian noise $p_0$) into the data distribution $p_1$, by constructing the vector field $u$ with a neural network, denoted by $u_t(\cdot, \theta) : \mathbb{R}^D \to \mathbb{R}^D$. This vector field $u_t$ is used to generate a time-dependent continuously differentiable map called *flow* $\phi_t(\cdot, \theta) : \mathbb{R}^D \to \mathbb{R}^D$ if the random variable generated with $X_t \sim \phi_t(x, \theta)$ satisfy Equation 1. A vector field $u_t$ is called to generate the flow $\phi_t$ if it satisfies the push-forward (i.e. (generalized) change of variables) equation

$$p_t = [\phi_t]_* p_0,$$
$$\text{where } [\phi_t]_* p_0(x) = p_0(\phi_t^{-1}(x)) \det \left[ \frac{\partial \phi_t^{-1}}{\partial x}(x) \right]. \tag{2}$$

However, constructing the CNF requires backpropagation of the entire adjoint equation with the NLL objective, which requires full forward simulation and gradient estimation of the vector field over the time domain $[0,1]$. The *flow matching* (Lipman et al., 2023; Liu et al., 2023a) algorithm overcomes this limit with a simulation-free (e.g., does not require a complete forward pass for training) algorithm by replacing this with the $\ell_2$-square objective

$$\mathcal{L}_{\text{FM}}(\theta) = \mathbb{E}_{t, p_1(x_1)} \left[ \|u_t(x, \theta) - v_t(x)\|_2^2 \right], \tag{3}$$

where $v_t(x)$ is true vector field that the neural network model $u_t$ aims to learn. However, the computational intractability of $v_t$ makes it difficult to directly learn from the true objective Equation 3. Lipman et al. (2023) found that by using the conditional flow matching objective instead,

$$\mathcal{L}_{\text{CFM}}(\theta) = \mathbb{E}_{t, p_1(x_1), p_t(x_t|x_1)} \left[ \|u_t(x_t, \theta) - v_t(x_t|x_1)\|_2^2 \right], \tag{4}$$

where $p_t(x_t|x_1)$ is the conditional probability density of noisy data $x_t$ given the data $x_1$, we can learn the flow matching model with conditional flow matching objective based on the proposition below.

---

**Algorithm 1** Training Sequential Rectified Flow for Generative Modeling

---

**Require:** Data distribution $\pi_{\text{data}}$, Noise distribution $\pi_{\text{noise}}$, data dimension $D$, Number of divisions $K$, Flow network model $v_\theta : \mathbb{R}^D \to \mathbb{R}^D$, Time division $\{t_i\}_{i=1}^{K-1}$, $0 = t_0 < t_1 < \cdots < t_{K-1} < t_K = 1$.
  **Stage 1. Pre-training rectified flow**
  **while** Not converged **do**
      Draw $(X_0, X_1) \sim \pi_{\text{data}} \times \pi_{\text{noise}}$, $t \in (0, 1)$.
      $X_t \leftarrow (1 - t)X_0 + tX_1$.
      Update $\theta$ to minimize $\mathbb{E}_{X_t, t}\left[\|v_\theta(X_t, t) - (X_1 - X_0)\|_2^2\right]$   (Learning the flow network)
  **end while**
  **Stage 2. Sequential reflow + Distillation with segmented time divisions**
  **while** Not converged **do**
      Sample $X_{t_i} = (1 - t_i)X_0 + t_i X_1$, $(X_0, X_1) \sim \pi_{\text{data}} \times \pi_{\text{noise}}$, $t_i, i \in \{1, 2, \cdots, K\}$
      Obtain $\hat{X}_{t-1}$ by running $\mathrm{d}X_t = v_\theta(X_t, t)\mathrm{d}t$ with an ODE solver from $t_i \to t_{i-1}$.
      **if** Reflow **then**
          $X_s \leftarrow (1 - r)\hat{X}_{t_{i-1}} + rX_{t_i}$, $s = t' + (t_i - t_{i-1})r$, $r \sim \mathcal{U}(0, 1)$
      **else if** Distill **then**
          $X_s \leftarrow X_{t_i}$, $s = t$
      **end if**
      Update $\theta$ to minimize $\mathbb{E}_{X_s, r}\left[\left\|v_\theta(X_s, s) - \frac{X_s - X_{t_{i-1}}}{t_i - t_{i-1}}\right\|_2^2\right]$   (Learning with sequential reflow)
  **end while**

---

**Proposition 2.1** (Equivalence of the FM and CFM objective). *The gradient of Equation 3 and Equation 4 is equal. That is, $\mathcal{L}_{\text{FM}}(\theta) = \mathcal{L}_{\text{CFM}}(\theta) + C$ for a constant $C$. The detailed proof is in Appendix C.*

A natural choice to construct flow between two distributions is to independently draw samples from two distributions and let the vector field at $t$ be the conditional Gaussian, centered at the linear interpolants of these samples. Liu et al. (2023a); Albergo & Vanden-Eijnden (2023) designed the source distribution $p_1$ and the conditional distribution as

$$p_1 \sim \pi_0 \times \pi_1, \quad v_t(x_t | x_0, x_1) \sim \mathcal{N}(tx_1 + (1 - t)x_0, \sigma^2) \tag{5}$$

that is, the source distribution is the product of two independent distributions $\pi_0$ and $\pi_1$, and the conditional vector field $v_t$ is the (stochastic) linear interpolation between the two conditional data drawn independently from two distributions. Then, the CFM objective becomes

$$\mathbb{E}_{t, \pi_0(x_0), \pi_1(x_1), n \sim \mathcal{N}(0, \sigma^2)}\left[\|u_t(x_t, \theta) - (tx_1 + (1 - t)x_0 + n)\|\right], \tag{6}$$

which is reduced to

$$\mathbb{E}_{t, \pi_0(x_0), \pi_1(x_1)}\left[\|u_t(x_t, \theta) - (tx_1 + (1 - t)x_0)\|\right] \tag{7}$$

when the conditional vector field is given deterministic, i.e., $\sigma = 0$, and corresponds to the rectified flow objective (Liu et al., 2023a;b).

## 3 SEQUENTIAL REFLOW

In this section, we propose our main method called *sequential reflow*, a training technique that can be unifiedly applied to continuous-time generative models, including flow-based models and diffusion models. We first introduce the overall method in § 3.1, and provide further discussions on our method in § 3.2. The sketch of our training algorithm is explained in Algorithm 1.

### 3.1 TIME-SEGMENTED RECTIFIED FLOW FOR GENERATIVE MODELING

**Constructing flow.** Following the widely used existing literature (Liu et al., 2023a;b; Albergo & Vanden-Eijnden, 2023; Albergo et al., 2023), we first construct the rectified flow that simulates the ODE with the conditional flow matching objective Equation 7 by first drawing $(x_0, x_1) \sim \pi_0 \times \pi_1$

from independent distributions. For generative modeling, for instance, we use $\pi_0 = \pi_{\text{data}}$ and $\pi_1 = \mathcal{N}(0, I^D)$, respectively. Then, the conditional vector field $v_t(x_t, t)$ yields the gap between the drawn data and noise sample. As we investigated in § 2, this conditional flow matching objective finally constructs the (marginal) flow $\psi_t$, which is a push-forward between the noise and data distributions. The **Stage 1** of Algorithm 1 summarizes the training procedure via this flow matching objective.

**Generating Joint Distribution.** Even though the vector field $v_t$ constructed by rectified flow is known to marginally converge to the true vector field $u_t$, this is intractable in large-scale datasets because of high training variance. Pooladian et al. (2023) provides the variance reduction argument that sheds light on using joint distribution as training data, as follows.

**Proposition 3.1** (Pooladian et al. (2023), Lemma 3.2). *Let $(x_0, x_1)$ be jointly drawn from the joint density $\pi = \pi_0 \times \pi_1$. And let the flow-matching objective from joint distribution be given as*

$$\mathcal{L}_{\text{JCFM}} = \mathbb{E}_{\pi(x_0, x_1)} \left[ \| u_t(x_t; \theta) - v_t(x_t | x_1) \|_2^2 \right]. \tag{8}$$

*Then the expectation of the total variance of the gradient at a fixed $(x, t)$ is bounded as*

$$\mathbb{E}_{t, p_t(x)} \left[ \text{Tr}(\text{Cov}_{p_t(x_1 | x)} \left( \nabla_\theta \| u_t(x; \theta) - v_t(x | x_1) \|_2^2 \right) \right] \leq \max_{t, x} \| \nabla_\theta u_t(x; \theta) \|_2^2 \mathcal{L}_{\text{JCFM}}. \tag{9}$$

This proposition implies that training from joint distributions, instead of the product of independent distributions, improves training stability and thus efficiently constructs the optimal transport (OT) mapping between two distributions.

Recently, Liu et al. (2023a) introduced the *reflow* algorithm to reduce training variance and straighten the trajectory by drawing the joint distribution by starting from noise and then simulating through the ODE solver to generate samples. The idea of reflow is simple: first sample noise from $\pi_1 = \mathcal{N}(0, I^D)$, then run the reverse-time ODE with existing ODE solvers, such as the Euler method, Runge-Kutta method, or Heun's method to obtain the output of the reverse-time ODE from $t = 1$ to $t = 0$.

**Main method: Sequential reflow (($K$)-SeqRF).** Even though the Reflow algorithm seeks the rectified path between distributions, the ODE solver, which is used to simulate the Reflow algorithm, always has inaccuracy because of the high curvature of the probability path. As the ODE solver always has inaccuracy within a single step, and this error accumulates over the running steps, the output of the reverse-time ODE from the initial noise does not correspond to the actual output of the continuous-time ODE, causing the *global truncation error*. The detailed interpretation and analysis of this phenomenon are further described in § 3.2. And we denote $K$-SeqRF as sequential reflow with $K$ intervals.

Our new method, called *sequential reflow (SeqRF)*, proposes a simple and efficient way to deal with this issue by dividing the time domain of the ODE into multiple steps. To be precise, let the entire time interval be $t \in [0, 1]$, where the $X_0 \sim \pi_0$ and $X_1 \sim \pi_1$. Then we divide the interval $[0, 1]$ into $K$ steps, as $0 = t_0 < t_1 < t_2 < \cdots < t_K = 1$. Then we draw the joint distribution by (1) random sampling the interval $[t_{i-1}, t_i]$ where $i \in [1 : K]$, (2) Drawing $(x_0, x_1)$ independently from $x_0 \in \pi_0$ and $x_1 \in \pi_1$, (3) Sampling the initial data at time $t_i$ as $X_{t_i} = t_i X_0 + (1 - t_i) X_1$, and finally (4) solve the ODE, beginning in $x_{t_i}$ at time $t_i$ until $t_{i-1}$.

**Distillation.** After training the probability path with joint distribution, we fine-tuned our model with distillation in order to move directly toward the whole time segment. That is, the number of function evaluations is the same as the number of time segments after distillation finishes. The distillation process is done as follows: we use the same generated dataset as joint distribution, but we fix the time to the starting point of the time segment instead of uniformly training from $t \in [0, 1]$. Then, by distilling the reflow model, we directly take advantage of the straightened probability flow of the previously trained SeqRF model. We later interpret the straightness of the flow in the SeqRF model in § 3.2 to delve more into this.

### 3.2 MAIN PROPERTIES OF SEQUENTIAL REFLOW (SEQRF).

In this section, we provide how the SeqRF algorithm provides a straighter flow. First, we show theoretically that SeqRF provably mitigates the global truncation error, which is an upper bound

of the global error of an ODE solver. After that, we empirically validate that SeqRF rectifies the flow, by showing that the vector field constructed by SeqRF is straighter in image datasets such as CIFAR-10 and CelebA.

### 3.2.1 FIXING THE GLOBAL TRUNCATION ERROR

Let the Initial Value Problem (IVP) on an ODE be given as follows:

$$\mathrm{d}X_t = v(X_t, t)\mathrm{d}t, \quad t \in [a, b], X_a = x(0). \tag{10}$$

Let the initial value $a$ and the terminal value $b$ be divided by $h$ equidistributed intervals, i.e., $t_i = a + \frac{i}{h}(b - a)$, and let the generalized *linear multistep method* which includes most of widely used ODE methods, solving this ODE IVP be given as

$$x(i+1) = \sum_{j=0}^{p} \alpha_j x(i-j) + h \sum_{j=-1}^{p} \beta_j v(x(i-j), t_{i-j}), \quad i \ge p. \tag{11}$$

Then the *global truncation error* of the ODE solver $\|X_b - x(t_h)\|$ is defined by the absolute difference between the true ODE solution and the output of the solver at the terminal value $b$. Then, Dahlquist (1963) provided that the upper bound of the global truncation error is as follows.

**Lemma 3.2** (Dahlquist Equivalence Theorem (Dahlquist, 1963))**.** *For a linear multistep method that is consistent with the* ODE *(10) where $v_t$ satisfy the Lipschitz condition and with fixed initial value $x_a$ at $t = a$, the global truncation error is $\mathcal{O}(h^r)$.*

With this lemma, we provably show that the sequential reflow algorithm reduces the global truncation error by the order of $K^{r-1}$, where $K$ is the number of intervals and $r$ is the order of the ODE solver. To be precise,

**Theorem 3.3** (The global truncation error of the Sequential reflow algorithm)**.** *Suppose that a linear multistep method (11) successfully simulates the* ODE *(10). And suppose that the Number of Function Evaluation (NFE) of the reflow and the sequential reflow algorithm is given the same as $\frac{b-a}{K}$. And let the global truncation error of the reflow algorithm and the sequential reflow be $\mathrm{GTE}_{\mathrm{RF}}$ and $\mathrm{GTE}_{\mathrm{SeqRF}}$, respectively. Let $K$ be the number of equidistributed intervals that segment $[a, b]$. And let (11) is of order $p$, i.e. the local truncation error is $\mathcal{O}(h^{p+1})$, then the following holds.*

$$\mathrm{GTE}_{\mathrm{SeqRF}} = \mathcal{O}\left(K^{1-r}\right) \mathrm{GTE}_{\mathrm{RF}} \tag{12}$$

*Proof.* Please refer to Appendix B. □

### 3.2.2 FLOW STRAIGHTENING EFFECT

Liu et al. (2023a) proposed a measure called *straightness* of a continuously differentiable process $\mathbf{Z} = \{Z_t\}_{t=0}^{1}$ to measure how straight the path generated by the ODE solver, as defined by

$$S(\mathbf{Z}) = \int_0^1 \left\| (Z_1 - Z_0) - \frac{\mathrm{d}}{\mathrm{d}t} Z_t \right\|^2 \mathrm{d}t, \tag{13}$$

where zero $S(\mathbf{Z})$ means the process is completely straight and the vector field is constant over time, which implies that the Euler solver can yield the exact sample in just a single function evaluation. We propose a metric called *sequential straightness* to validate how straight the sequential flow (e.g., how much the truncation error is mitigated.) of a collection of time-segmented processes $\mathcal{Z} = \{\mathbf{Z}^i\}_{i=1}^{K}$ as

$$S_{\mathrm{seq}}(\mathcal{Z}) = \sum_{i=1}^{K} \int_{t_{i-1}}^{t_i} \left\| \frac{Z_{t_i}^i - Z_{t_{i-1}}^i}{t_i - t_{i-1}} - \frac{\mathrm{d}}{\mathrm{d}t} Z_t^i \right\|^2 \mathrm{d}t, \tag{14}$$

where zero $S_{\mathrm{seq}}(\mathcal{Z})$ denotes that each time-segmented process is completely straight.

Figure 3 demonstrates the sequential straightness of the rectified flow models trained on CIFAR-10 and CelebA datasets with respect to the number of segments, where the number of reflow pairs is fixed to $50,000$ for both datasets and the pre-trained rectified flow is fine-tuned with those reflow

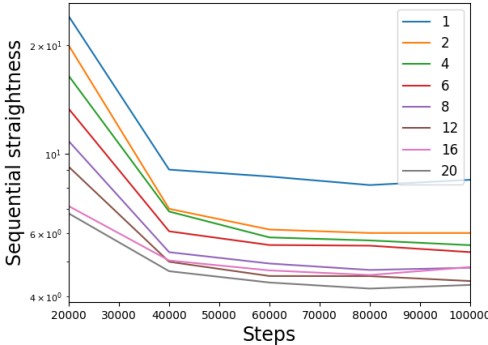 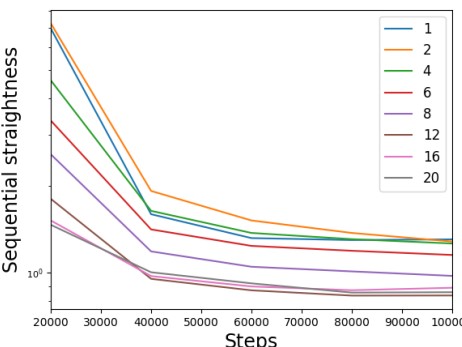

Figure 3: The number of steps vs. the sequential straightness with respect to the number of equidistributed segments within $t \in [0, 1]$ over the number of sampling steps. The legends denote the number of segments of sequential reflow; $N = 1$ denotes the naïve reflow. Having lower sequential straightness stands for a better flow straightening effect. **Left**: CelebA, **Right**: CIFAR-10 dataset.

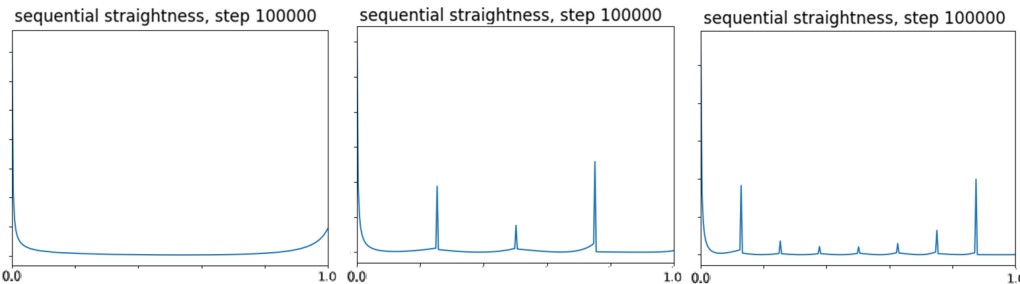

Figure 4: The sequential straightness over time in training SeqRF with CIFAR-10 dataset in 100,000 steps, in {1,4,8}-SeqRF, 4-SeqRF, 8-SeqRF, from left to right.

pairs by $100,000$ steps when the batch size is $128$. The sequential straightness lowers in both datasets as the we train with finer intervals, implying that we achieved a superior straightening performance by using the SeqRF method.

Figure 4 demonstrates the mean sequential straightness with respect to time when we train SeqRF with CIFAR-10 dataset. We can observe that the sequential straightness at $t = 0$ (near data) or $t = 1$ (near noise) is greater than other $t$, which implies that straightening the ODE trajectory is more difficult in near-data or near-noise regime, which is an analogous observation as Choi et al. (2021).

Moreover, this figure implies the evidence that distillation improves the performance in a large margin, as introduced in Figure 1. The sequential straightness is shown to spike at the boundary regions between adjacent intervals, since in the boundary point we learn the vector field of both facing regions, that may cause instability in training the right vector field. The distillation scheme we have proposed focuses on the spiking boundary points, which will mitigate the spiking phenomenon.

For details in the time-dependent sequential correctness and implementation details, refer to Appendix D.

## 4 RELATED WORK

As a problem of designing ODEs as probability path, Chen et al. (2018) has opened up a new way by showing that an ODE can be learned with neural network objectives, and Grathwohl et al. (2019) further applied this for the generative models. Generating the invertible flow has been also achieved as architecture design (Kingma & Dhariwal, 2018) and from autoregressive model (Papamakarios

et al., 2017; Huang et al., 2018). Those early methods, however, are computationally inefficient, had memory issue, or not competitive in terms of synthesis quality in generative model literature.

As a family of continuous-time generative modeling by learning dynamics, Sohl-Dickstein et al. (2015) interpreted the generative modeling as the construction of the vector field of the stochastic processes. Later on, Ho et al. (2020); Song & Ermon (2019); Song et al. (2021b) proposed efficient techniques for training this by interpreting the reverse stochastic process as the denoising model. According to the diffusion models trained on SDEs, Song et al. (2021a) proposed an efficient technique by sampling from a generalized non-Gaussian process, which results in a fast deterministic sampling speed. Further, Kingma et al. (2021); Huang et al. (2021) unified the framework of continuous-time models, including diffusion model and flow-based generative models.

Recently, straightening the continuous flow by mitigating the curvature of the probability path has been considered by considering the optimal transport regularization by introducing CNF norms (Finlay et al., 2020; Onken et al., 2021; Bhaskar et al., 2022) for training neural ODE, and Kelly et al. (2020) learned the straightened neural ODEs by learning to decrease the surrogate higher-order derivatives. Lee et al. (2023) showed that minimizing the curvature in flow matching is equivalent to the $\beta$-VAE in variational inference. As a

Table 1: CIFAR-10 and CelebA Generation Results on Flow Matching Methods. The superscript[†] denotes the re-implementation based on the authors' proposed hyperparameters using `JAX/Flax`.

| **CIFAR-10** | | | | |
|---|---|---|---|---|
| Method | NFE ($\downarrow$) | IS ($\uparrow$) | FID ($\downarrow$) | Solver |
| 1-Rectified Flow | 127 | 9.60 | 2.58 | RK45 |
| (Liu et al., 2023a) | 1 | 1.13 | 378 | Euler |
| (+Distill) | 1 | 9.08 | 6.18 | Euler |
| 2-Rectified Flow | 110 | 9.24 | 3.36 | RK45 |
| | 1 | 8.47 | 12.21 | Euler |
| (+Distill) | 1 | 8.79 | 4.85 | Euler |
| 3-Rectified Flow | 104 | 9.01 | 3.96 | RK45 |
| | 1 | 8.47 | 8.15 | Euler |
| (+Distill) | 1 | 8.79 | 5.21 | Euler |
| Conditional FM | 183 | - | 8.06 | |
| I-CFM | 50 | - | 10.68 | Euler |
| OT-CFM | 10 | - | 20.86 | Euler |
| DDPM (Ho et al., 2020) | 1000 | | 3.16 | |
| Analytic-DDPM (Bao et al., 2022) | 50 | | 7.34 | |
| Analytic-DDIM (Bao et al., 2022) | 50 | | 4.28 | |
| EDM (Karras et al., 2022) | 35 | | 1.97 | |
| 1-Rectified Flow[†] | 1000 | - | 3.03 | Euler |
| 2-Rectified Flow[†] | 100 | - | 4.54 | Euler |
| (+Distill)[†] | 1 | - | 4.85 | Euler |
| 4-SeqRF (+Distill) (Ours) | 4 | 9.09 | 4.75 | Euler |
| 6-SeqRF (+Distill) (Ours) | 6 | 9.17 | 4.29 | Euler |
| 8-SeqRF (+Distill) (Ours) | 8 | **9.32** | **3.97** | Euler |
| **CelebA** | | | | |
| 1-Rectified Flow[†] | 1000 | - | 2.37 | Euler |
| 2-Rectified Flow[†] | 100 | - | 4.35 | Euler |
| (+Distill)[†] | 1 | - | 6.22 | Euler |
| 2-SeqRF (+Distill) (Ours) | 2 | - | 4.95 | Euler |
| 4-SeqRF (+Distill) (Ours) | 4 | - | **3.67** | Euler |

generalization of finding optimal transport interpolant, Albergo et al. (2023); Albergo & Vanden-Eijnden (2023) proposed stochastic interpolant, which unifies the flows and diffusions by showing that the probability density of the noisy interpolant between two distributions satisfies the Fokker-Planck equations of an existing diffusion model.

Our work can also be interpreted to match the noise and data distribution pairs to make training more efficient by reducing training variance. Pooladian et al. (2023) used the minibatch coupling to generate joint distributions in a simulation-free manner by constructing a doubly stochastic matrix with transition probabilities and obtaining coupling from this matrix. Even though this generates one-to-one matching between noise and images, this is numerically intractable when the number of minibatch increases. Tong et al. (2023b) also introduced the concept of minibatch optimal transport by finding optimal coupling within a given minibatch. Tong et al. (2023a) further expanded this approach to generalized flow (e.g., Schrödinger bridge) in terms of stochastic dynamics.

## 5 Experiments

In this section, we empirically evaluate SeqRF, compared to other generative models especially with diffusion and flow-based models in CIFAR-10 and CelebA datasets. We used the `NCSN++` architecture based on the repository of Song et al. (2021b), implemented with `JAX/Flax` packages on TPU-v2-8 (CIFAR-10) and TPU-v3-8 (CelebA) nodes. In CIFAR-10 and CelebA dataset, we used 1M and 200k reflow datasets for generating reflow models or SeqRF models, and 100k reflow datasets for distillation in both cases. The whole experimental details are described in Appendix A.

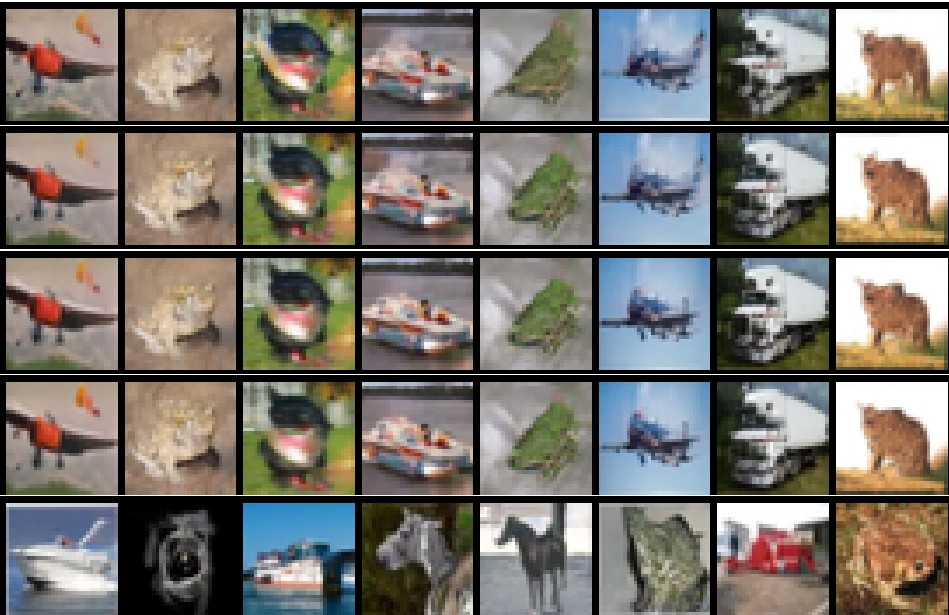

Figure 5: Non-curated CIFAR-10 image synthesis result with 2-SeqRF after distillation. From **up** to **down**: {1, 2, 4, 6, 8}-step Euler solver, each with FID score of 11.89, 6.97, 4.75, 4.29, 3.97 with our own implementation.

## 5.1 IMAGE GENERATION RESULT

We demonstrate our image synthesis quality in Table 1 in CIFAR-10 and CelebA domain, compared to existing flow matching methods including rectified flow (Liu et al., 2023a), conditional flow matching (Lipman et al., 2023), I-CFM and OT-CFM (Tong et al., 2023b). In case of rectified flow, we re-implemented the method using JAX/Flax packages. Both in CIFAR-10 and CelebA dataset, we achieved superior performance compared to other flow-based models, also achieving high sampling speed with less than ten function evaluations. As a comparing model to ours, the rectified Reflow model (2-RF) has its performance in very long-NFE region as 4.54 in CIFAR-10 dataset and 4.35 in CelebA datasets, respectively. As the distilled performance cannot surpass this extreme point, the 2-Reflow model cannot reach our FID of 3.97 in CIFAR-10 and 3.67 in CelebA dataset with previous methods. For additional images generated by SeqRF, please refer to Appendix E.

## 6 CONCLUSION AND DISCUSSION

We introduce sequential reflow, a straightforward and effective technique for rectifying flow-based generative models. This method initially subdivides the time domain into multiple segments and subsequently generates a joint distribution by traversing over partial time domains. Through the division of the time domain, we successfully alleviate the global truncation error associated with the ODE solver. Consequently, this process yields a straighter path, thereby enhancing the efficiency and speed of the sampling procedure.

While our method has been validated for addressing truncation errors within the context of a specific flow matching method, namely rectified flow, its applicability extends beyond this specific framework. Our proposed sequential reflow technique can be readily applied to diverse frameworks utilized for constructing ODEs. Moreover, its utility is not limited to the domain of flow matching; rather, it holds potential for broader applications in tasks such as image-to-image translation and domain adaptation. The versatility of our method underscores its potential to enhance the stability and efficiency of various ODE-based models and applications beyond the specific instance studied in this work.

## 7 ETHICS STATEMENT

This is the work in the field of generative model algorithm, which is not directly involved in an ethical issue.

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

## A  Experimental Details

### A.1  Dataset Description

**CIFAR-10**  The CIFAR-10 dataset is the image dataset that consists of 10 classes of typical real-world objects with $50,000$ training images and $10,000$ test images with $32 \times 32$ resolution. In our experiments, we did not use the image labels and constructed unconditional generative models.

**CelebA**  The CelebA dataset is the face dataset that consists of $202,599$ training images from $10,177$ celebrities with 40 binary attribute labels, with size $178 \times 218$. In our experiment, we resized and cropped the image to $64 \times 64$ resolution to unify the input shape of the generative models. Also, we did not use the attribute labels and constructed unconditional generative models.

**LSUN-Church**  The Large-Scale Scene UNderstanding (LSUN)-Church(-Outdoor) dataset is the dataset that consists of the church images as well as the background which surrounds them. This data consists of $126,227$ images with $256 \times 256$ resolution.

### A.2  Details on Training Hyperparameters

Table 2: The hyperparameter used to train our model.

|  | CIFAR-10 | CelebA | LSUN-Church |
|---|---|---|---|
| Channels | 128 | 128 | 128 |
| Depth | 3 | 3 | 4 |
| Channel multiplier | (1, 2, 2) | (1, 2, 2) | (1, 2, 2, 4) |
| Heads | 4 | 4 | 4 |
| Attention resolution | 16 | 16 | 16 |
| Dropout | 0.15 | 0.1 | 0.1 |
| Effective batch size | 128 | 128 | 32 |
| TPUs | v2-8 | v3-8 | v3-8 |
| Steps | 100000 | 100000 | 200000 |
| Reflow images | 1M | 200k | 500k |
| Learning rate (Adam) | $2e-3$ | $2e-3$ | $5e-4$ |

We report the hyperparmeters that we used for training in Table 2. We used 32-bit precision floating number for training all datasets. For CIFAR-10 and CelebA datasets, we used Adam optimizer with $\beta_1 = 0.5$ and $\beta_2 = 0.9$. The architectural hyperpameters are specified in (Table).

**EMA rate**  Since the flow matching and rectified flow is much harder to converge than the conventional diffusion model because of matching pairs of the independent distributions, we tuned the EMA rate in two ways:

- We have increased the EMA rate to $0.999999$, higher than other models. (Table) shows the ablation study on the EMA rate of the image synthesis quality from CIFAR-10 datasets from baseline 1-rectified flow model for a 1000-step Euler solver.

- **Warm-up training policy.**  Fixing the EMA rate high makes converging the model almost impossible because of the extremely high momentum. So, we introduce the warmup phase with respect to the training step as

$$\texttt{EMA\_rate=min((1 + step) / (10 + step), decay)} \qquad (15)$$

  where `decay` and `step` are the given EMA rate and the training steps, respectively. Introducing this warmup phase further improved the FID of our implementation in 1.3M steps and 128 batch size to **3.03** in CIFAR-10 dataset generation using a 1,000-step Euler solver.

Table 3: The performance of the 1-rectified flow model, trained with batch size 128, 1.3M steps and different EMA rates. The model does not converge with EMA rate 0.999999.

| EMA rate | 0.999 | 0.9999 | 0.99999 | 0.999999 | Warm-up |
|----------|-------|--------|---------|----------|---------|
| FID      | 5.78  | 4.77   | 3.91    | -        | **3.03** |

## B  ON THE GLOBAL TRUNCATION ERROR OF THE LINEAR MULTISTEP ODE SOLVER

Let the linear multistep method is defined as

$$y_{n+1} = \sum_{j=0}^{p} \alpha_j x(i-j) + h \sum_{j=-1}^{p} \beta_j v_t(x(i-j), t_{i-j}), i \geq p \tag{16}$$

where the interval of a single step is $h$, $t_i = t_0 + hi$ stands for $i$-th time step, and $v_t$ is the vector field, or the drift function. Let the linear multistep method have the convergence order of $r$. Then according to the *Dahlquist equivalence theorem*, the global truncation error of the ODE solver has the order of $\mathcal{O}(h^r)$. In case of the sequential reflow algorithm with the same NFE, the single interval of a single step is given as $\frac{h}{K}$ where $K$ is the number of equidistribted intervals that divide $[a, b]$ into time segments. Then the global truncation error of the ODE solver of the sequential reflow algorithm is of $\mathcal{O}\left(\left(\frac{h}{K}\right)^r\right)$. As we have $K$ segments, the order of the global truncation error is $K \times \mathcal{O}\left(\left(\frac{h}{K}\right)^r\right) = K^{1-r}\mathcal{O}(h^r)$. $\qquad\square$

## C  EQUIVALENCE OF FLOW MATCHING AND CONDITIONAL FLOW MATCHING

In this section, we recap the equivalence of gradients of flow matching objective Equation 3 and conditional flow matching objective Equation 4. For further information, refer to Lipman et al. (2023), Theorem 2.

**Proposition C.1** (Equivalence of Equation 3 and Equation 4). *Let $p_t(x) > 0, \forall x \in \mathbb{R}^D$ and $t \in [0, 1]$, where $p_t$ is the marginal probability path over $x$. Then, $\nabla_\theta \mathcal{L}_{FM}(\theta) = \nabla_\theta \mathcal{L}_{CFM}(\theta)$.*

*Proof.* First, we remind the two equations.

$$\mathcal{L}_{\text{FM}}(\theta) = \mathbb{E}_{t,p_1(x_1)} \left[ \|u_t(x, \theta) - v_t(x)\|_2^2 \right]$$
$$\mathcal{L}_{\text{CFM}}(\theta) = \mathbb{E}_{t,p_1(x_1),p_t(x_t|x_1)} \left[ \|u_t(x_t, \theta) - v_t(x_t|x_1)\|_2^2 \right]. \tag{17}$$

From direct computation, we have

$$\|u_t(x, \theta) - v_t(x)\|_2^2 = \|u_t(x, \theta)\|_2^2 - 2u_t(x) \cdot v_t(x) + \|v_t(x)\|_2^2$$
$$\|u_t(x, \theta) - v_t(x)\|_2^2 = \|u_t(x, \theta)\|_2^2 - 2u_t(x) \cdot v_t(x|x_1) + \|v_t(x|x_1)\|_2^2. \tag{18}$$

Since $\mathbb{E}_{p_t(x)} \left[ \|u_t(x, \theta)\|_2^2 \right] = \mathbb{E}_{p_t(x|x_1)q(x_1)} \left[ \|v(t)\|_2^2 \right]$, the first term at the right-hand side is removed. Since $v_t(x)$ and $v_t(x|x_1)$ is an analytically calculated form which is independent of $\theta$, the only remaining term is the dot products. Then

$$\mathbb{E}_{p_t(x)} \left[ u_t(x) \cdot v_t(x) \right] = \int p_t(x)(u_t(x) \cdot v_t(x)) \mathrm{d}x$$
$$= \int p_t(x) \left( u_t(x) \cdot \int \frac{v_t(x|x_1)p_t(x|x_1)q(x_1)}{p_t(x)} \mathrm{d}x_1 \right) \mathrm{d}x \tag{19}$$
$$= \int \int u_t(x)q(x_1) \left( u_t(x) \cdot v_t(x|x_1) \right) \mathrm{d}x_1 \mathrm{d}x$$
$$= \mathbb{E}_{q(x_1)u_t(x)} \left[ u_t(x)v_t(x|x_1) \right]$$

where the change of integration is possible since the flow is a diffeomorphism. $\qquad\square$

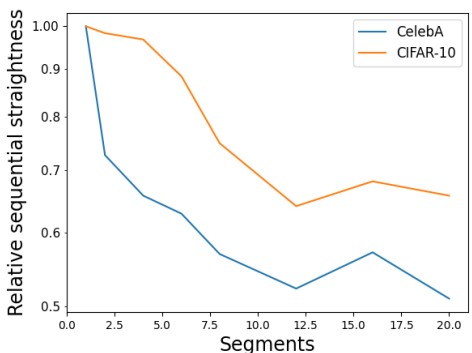 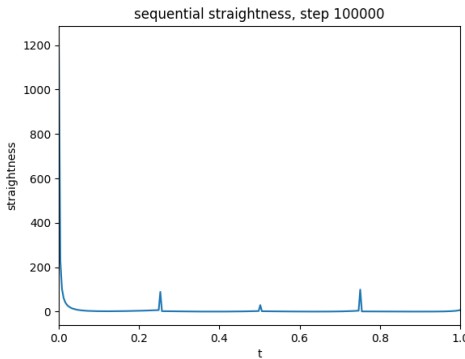

Figure 6: More results on sequential straightness. From **upper left** to **lower right**: the relative sequential straightness at 100k steps, compared to the original reflow scheme.

# D    SEQUENTIAL STRAIGHTNESS

## D.1    IMPLEMENTATION DETAILS

We measured the sequential straightness

$$S_{\text{seq}}(\mathcal{Z}) = \sum_{i=1}^{K} \int_{t_{i-1}}^{t_i} \left\| \frac{Z_{t_i}^i - Z_{t_{i-1}}^i}{t_i - t_{i-1}} - \frac{\mathrm{d}}{\mathrm{d}t} Z_t^i \right\|^2 \mathrm{d}t \tag{20}$$

by the following procedure.

(1) Random draw the time interval $[t_{i-1}, t_i]$ with $i \in [1 : K]$, as in the SeqRF algorithm.

(2) Sample the oracle input at time $t_i$ as $Z_{t_i} = (1 - t_i)X_0 + t_iX_1$.

(3) Then run the ODE solver to collect all sample within $[t_{i-1}, t_i]$.

(4) $Z_{t_{i-1}} = \hat{X}_{t_{i-1}}$, that is, the terminal value of the reverse-time ODE solver at time $t_{i-1}$.

(5) The vector field is calculated as $\frac{Z_{t_i}}{-} Z_{t_{i-1}} t_i - t_{i-1}$, and the derivative is defined as the difference between the values of two adjacent ODE solver timesteps, divided by the size of timesteps.

## D.2    MORE RESULTS ON SEQUENTIAL STRAIGHTNESS

Figure 6 demonstrates additional results on sequential straightness.

# E   ADDITIONAL FIGURES

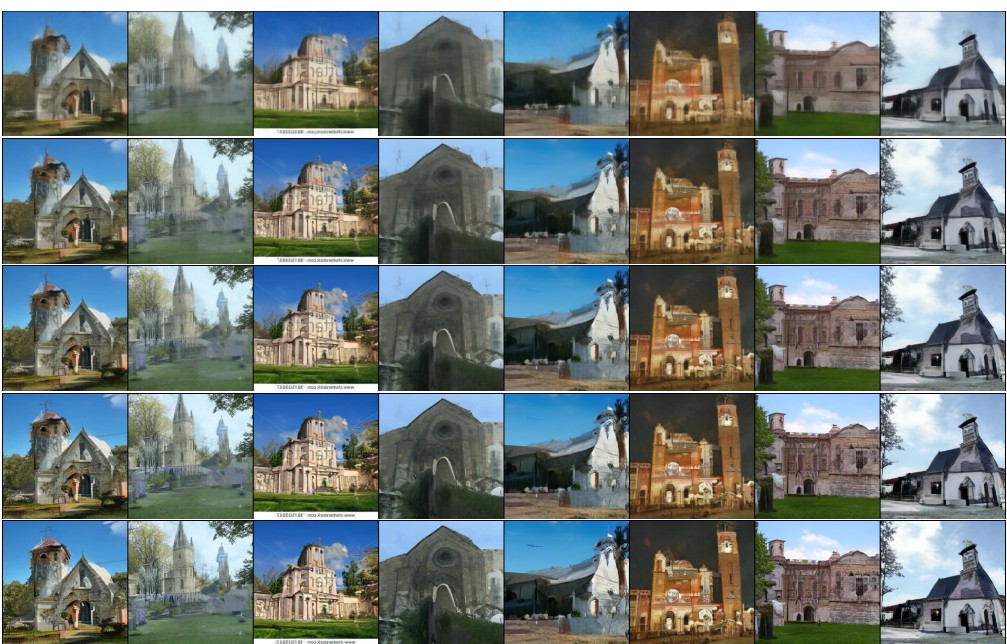

Figure 7: Non-curated LSUN-Church-256 image synthesis result with 2-SeqRF, that starts from the same random noise. From **up** to **down**: {1, 2, 5, 10, 25}-step Euler solver.

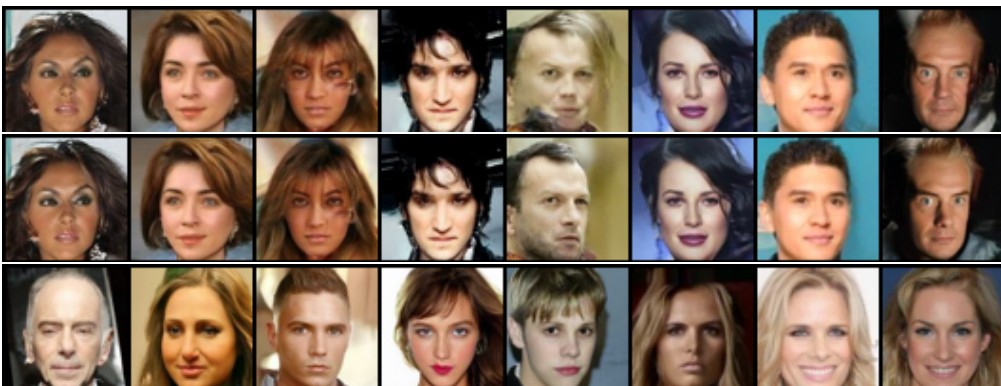

Figure 8: Non-curated CelebA images with 2-SeqRF after distillation. From **up** to **down**: {1, 2, 4}-step Euler solver, each have FID score of 6.22, 4.95, 4.67.

