# OpenReview forum: "Sequential Flow Straightening for Generative Modeling"
_ICLR.cc/2024/Conference — ICLR 2024 Conference Withdrawn Submission_

### Official Review · Reviewer_dQL3 · 2023-10-27

**Soundness:** 2 fair
**Presentation:** 1 poor
**Contribution:** 2 fair
**Rating:** 3
**Confidence:** 4

**Summary:**

This paper presents an imrpovement of the fine-tuning process for pretrained flow matching models. Flow matching [Lipman et al. 2023, Liu et al. 2023, Albergo & Vanden-Eijnden 2023] is a recently introduced type of generative models that are closely connected with score-based diffusion models, but are both empirically and theoretically demonstrated to be more beneficial to train and sample from. The improved fine-tuning is called  **Sequential Flow Straightening**, inspired by the reflow technique in [Liu et al. 2023]. Both take a pretrained flow matching model -- a velocity network, functioning very similar to score network -- then retrain it with some additional specific constraints. The authors of this paper argue that the original reflow technique from [Liu et al. 2023] will accumulate high global truncation error (which is the discretization error of nummerical ODE solver for sampling process). They therefore proposed a simple fix by doing reflow not on the whole trajectory [0, 1], but on smaller augmented paths that are arguably easier to perform reflow.

Xingchao Liu, Chengyue Gong, and Qiang Liu. Flow straight and fast: Learning to generate and transfer data with rectified flow. In International Conference on Learning Representations (ICLR), 2023a.
Yaron Lipman, Ricky T. Q. Chen, Heli Ben-Hamu, Maximilian Nickel, and Matthew Le. Flow matching for generative modeling. In International Conference on Learning Representations (ICLR), 2023
Michael S. Albergo and Eric Vanden-Eijnden. Building normalizing flows with stochastic interpolants. In International Conference on Learning Representations (ICLR), 2023.

**Strengths:**

* Sequential reflow deliver strong empirical performance,at least on the CIFAR10 and CelebA 64x64 on somewhat simple improvement of the reflow process.

**Weaknesses:**

* The biggest is presentation and writing of the paper. The writing feels rushed, many places are verborsed to read. Notations are inconsistent which are confusing. I will list here only some of the examples. In the background section (section 2) the authors used $x_1 \sim \pi_1$ to denote data and $x_0 \sim \pi_0$ to denote noises and their respective distribtions, but later on flip the notation. There is a notation $t'$ in Algorithm 1 which I do not understand where it comes from. The same applied to the quantity $r$ in Lemma 3.2, which is defined before and not after the result. Some part the random variable is denoted $X_t$, some $x(t)$, etc.

* The main method (the only contribution of this paper) consists of only two small paragraphs in the middle of page 5. There should be more elaboration, as I do not see a clear distinction between sequential reflow and the original reflow: are they different in flow matching loss function as well, or only sampling of the trajectory part? Most other parts are just reciting existing works' results, for example I do not understand the purpose of adding Proposition 3.1, which is cited from other paper and for me do not add anything to the analysis in section 3. This leaves a question on the contribution of the paper.

* Questionable benchmarks: I think the benchmark is a bit unfair as the authors do not take pretrained models and numbers from Rectified Flow paper (available freely online with CIFAR10 and CelebA-HQ at https://github.com/gnobitab/RectifiedFlow/), but instead reimplemented the methods in Jax, and do not provide their code in the Supplmentary Material. Using the same pretrained models (only differs in retraining method) with Rectified Flow paper would ensure a better comparison. in my opinion. What is more, the authors do not explicitly stated this in the main text, but instead of using CelebA with more standard resized 256x256 resolution, they trained theirs with 64x64 resolutions (only mentioned in the Appendix). This leads to a better, but misleading FID for the CelebA dataset.

**Questions:**

See the weakness section.

---

### Official Review · Reviewer_C8vt · 2023-10-28

**Soundness:** 3 good
**Presentation:** 3 good
**Contribution:** 2 fair
**Rating:** 3
**Confidence:** 4

**Summary:**

The paper presents a retraining technique for pre-trained rectified flow models. The proposed method aims to generate images of acceptable quality with a small number of function evaluations.

**Strengths:**

- The paper is easy to read and follow.

**Weaknesses:**

- My primary concern pertains to the novelty of the paper. In [1],  Liu et al. (2023) not only introduced the concept of a k-rectified flow but also proposed a k-step image generator, as discussed in Appendix A. This approach bears a similarity to your proposal; however, a key distinction arises in their implementation, where $X_s = X_1$ and $X_{t_i - 1} = X_0$. I train these k-step models, using the 1-rectified model from their GitHub repository, for various values of k, specifically 4, 6, and 8. The results obtained from these models closely resemble those achieved with 4-SeqRF (+Distill), 6-SeqRF (+Distill), and 8-SeqRF (+Distill) on Cifar10. Thus, I would strongly recommend that you consider comparing your method with this approach.

- What is the used ODE solver to calculate "$\hat{X}_{t-1}$" in Algorithm 1. A relatively precise ODE solver takes lots of time, being a bottleneck in the training process. Is it possible to precompute "\hat{X}_{t-1}"?

**Questions:**

- Can you provide the FID score of your method on LSUN-CHURCH?
- In the k-step image generator from [1], they select $t_0, t_1, t_2, \dots,t_K$ uniformly. How do you choose $t_0, t_1, t_2, \dots ,t_K$ ?
- In Fig. 5 and Fig. 8, the last row seems to be not consistent with other rows. Does a column contain images generated from the same noise?

**References**:

[1] Liu, Xingchao, Chengyue Gong, and Qiang Liu. "Flow straight and fast: Learning to generate and transfer data with rectified flow. International Conference on Learning Representations, 2023.

---

### Official Review · Reviewer_NVk8 · 2023-10-31

**Soundness:** 3 good
**Presentation:** 3 good
**Contribution:** 3 good
**Rating:** 6
**Confidence:** 3

**Summary:**

The goal of this paper is to address the challenge of slow sampling speed of diffusion and flow-based models with a novel method called sequential reflow. It applies the reflow algorithm from Liu et al in time-segments to reduce the accumulating error introduced by the ODE solver, resulting in 'straighter' flows within the segments. As a result, distilling the individual segments is more accurate, and the resulting segmented flow improves over standard reflow distilled flows at the cost of a slight increase in the number of function evaluations. The method is evaluated on CIFAR-10 and Celeb-A.

**Strengths:**

The paper's goal and contributions are clear, and the experiments suitable to validate the claims. The idea of applying reflow in segments to counter the accumulating error of the ODE solver is interesting, and the results are valuable to share with the community.

**Weaknesses:**

- Evaluation of the method is limited to the rectified flow model. To back up the claim of the general applicability, at least another method should be studied.
- It would be helpful to address the tradeoff between the compute saved by the reduced NFE and the compute required to run the reflow and distillation till convergence. When does it become worthwhile to apply this method?
- Even though the distilled reflow from Liu is designed for a single NFE, Table 1 should evaluate the competing methods for NFE's of 4,6,8 as well, to compare on an iso-NFE basis.

**Questions:**

1. Are there any limitations or potential drawbacks to the sequential reflow method when applied to other continuous-time generative models?
2. How can one determine the optimal number of segments of the time domain?
3. I don't follow the argument in the discussion of figure 4: If sequential straightness is greater near t=0 and t=1, why is straightening "more difficult" in these areas?
4. Figure 4 indicates that the sequential straightness is distributed around the endpoints of the trajectory. Does this motivate a non-linearly spaced segmentation scheme?

Nits:
1. Figure 1 would benefit from a legend, the line labels are not clear.
2. Figure 1's caption refers to SeqRef, but this term has not been introduced yet.